# Spontaneous Bacterial Peritonitis in Decompensated Liver Cirrhosis—A Literature Review

Chien-Hao Huang [1,2,*], Chen-Hung Lee [2,3] and Ching Chang [1]

1 Division of Hepatology, Department of Gastroenterology and Hepatology, Chang-Gung Memorial Hospital, Linkou Medical Center, Taoyuan 333, Taiwan
2 College of Medicine, Chang-Gung University, Taoyuan 333, Taiwan
3 Division of Cardiology, Department of Internal Medicine, Chang Gung Memorial Hospital-Linkou, Taoyuan 333, Taiwan
* Correspondence: huangchianhou@gmail.com; Tel.: +886-3-3281200 (ext. 8107)

**Abstract:** Background: Spontaneous bacterial peritonitis (SBP) is defined as a bacterial infection of the ascitic fluid without a surgically treatable intra-abdominal infection source. SBP is a common, severe complication in cirrhosis patients with ascites, and if left untreated, in-hospital mortality may exceed 90%. However, the incidence of SBP has been lowered to approx. 20% through early diagnosis and antibiotic therapy. Clinical awareness, prompt diagnosis, and immediate treatment are advised when caring for these patients to reduce mortality and morbidity. Aim: To discuss important issues comprising types of SBP, pathogenesis, bacteriology, including the emergence of multidrug-resistant (MDR) microorganisms, prompt diagnosis, risk factors, prognosis, treatment strategies, as well as recurrence prevention through antibiotic prophylaxis until liver transplantation and future trends in treating and preventing SBP in detail. Methods: This article is a literature review and appraisal of guidelines, randomized controlled trials, meta-analyses, and other review articles found on PubMed from between 1977 and 2022. Results: There are three types of SBP. Bacterial translocation from GI tract is the most common source of SBP. Therefore, two thirds of SBP cases were caused by Gram-negative bacilli, of which *Escherichia coli* is the most frequently isolated pathogen. However, a trend of Gram-positive cocci associated SBP has been demonstrated in recent years, possibly related to more invasive procedures and long-term quinolone prophylaxis. A diagnostic paracentesis should be performed in all patients with cirrhosis and ascites who require emergency room care or hospitalization, who demonstrate or report consistent signs/symptoms in order to confirm evidence of SBP. Distinguishing SBP from secondary bacterial peritonitis is essential because the conditions require different therapeutic strategies. The standard treatment for SBP is prompt broad-spectrum antibiotic administration and should be tailored according to community-acquired SBP, healthcare-associated or nosocomial SBP infections and local resistance profile. Albumin supplementation, especially in patients with renal impairment, is also beneficial. Selective intestinal decontamination is associated with a reduced risk of bacterial infection and mortality in high-risk group. Conclusions: The standard treatment for SBP is prompt broad-spectrum antibiotic administration and should be tailored according to community-acquired SBP, healthcare-associated or nosocomial SBP infections and local resistance profile. Since the one-year overall mortality rates for SBP range from 53.9 to 78%, liver transplantation should be seriously considered for SBP survivors who are good candidates for transplantation. Further development of non-antibiotic strategies based on pathogenic mechanisms are also urgently needed.

**Keywords:** spontaneous bacterial peritonitis; liver cirrhosis; culture-negative neutrophilic ascites; monomicrobial non-neutrocytic bacterascites; bacterial translocation; ascites fluid; gram-negative bacilli; gram-positive cocci; antibiotic prophylaxis

## 1. Introduction

Cirrhotic patients have an altered defense against bacteria associated with reduced bacterial clearance [1]. This immune defect facilitates bacterial translocation induced by increased intestinal permeability and gut bacterial overgrowth [2]. Therefore, bacterial infection is either present on admission or develops during hospitalization in about 30% of patients with cirrhosis [3], and the most common form of these infections is spontaneous bacterial peritonitis (SBP) [3].

SBP is a severe complication in cirrhosis patients with ascites [4]. Ascites is mainly transudative fluid with poor opsonic activity, which provides a favorable environment for growth of bacteria. The prevalence of SBP is 1.5–3.5% among outpatients and 10–30% among hospitalized patients [5]. When first reported, in-hospital mortality from an episode of SBP exceeded 90%; however, this rate has been lowered to approximately 20% through early diagnosis and prompt antibiotic therapy [6].

SBP is diagnosed upon positive ascites culture and/or absolute neutrophil count (polymorphonuclear cell or PMN) within ascites fluid (AF) of $\geq$250 cells/mm$^3$ [7,8]. Diagnosis is distinct from secondary peritonitis and hence is made in the absence of an intra-abdominal source of infection or other causes of an elevated ascites neutrophil count, such as hemorrhage, pancreatitis, peritoneal tuberculosis, and carcinomatosis [7], or an evident intra-abdominal, surgically treatable source [9,10].

Clinical awareness, prompt diagnosis, and appropriate treatment remain the most important tools for clinicians when caring for patients who experience SBP [11]. Prevention of SBP recurrence through antibiotic prophylaxis is another important consideration that will be discussed in the following literature review.

## 2. Materials and Methods

This article aims to help clinicians and other healthcare professionals in reviewing, studying, and assisting with the management of SBP patients. We have reviewed scientific literature found in the PubMed database published between 1977 and 2022. Key terms in our search include SBP, bacterial peritonitis, antibiotics, antibiotic resistance, ascites, paracentesis, microbiology, treatment, and prophylaxis. Randomized controlled trials and meta-analyses conducted for the treatment of SBP were also identified.

## 3. Types of SBP

There are three types of SBP (Table 1): (1) classic SBP; polymorphonuclear cell (PMN, also referred to as neutrophils) count in ascitic fluid is $\geq$250 cells/mm$^3$ and positive ascites culture; (2) culture-negative neutrophilic ascites (CNNA), ascites with a PMN count of $\geq$250 cells/mm$^3$ but with negative ascitic fluid culture; (3) monomicrobial non-neutrocytic bacterascites (MNB), PMN not elevated in ascites, but culture is positive.

**Table 1.** Three types of SBP.

| Ascites Fluid | Classic SBP | CNNA [1] | MNB [2] |
|---|---|---|---|
| PMN count (cells/mm$^3$) | $\geq$250 | $\geq$250 | <250 |
| Ascites culture | positive | negative | positive |

[1] CNNA: culture-negative neutrophilic ascites; [2] MNB: monomicrobial non-neutrocytic bacterascites.

The prevalence of classic SBP was high until 2007, with almost 80% of SBP culture-positive during this period [12]. However, since the adoption of antibiotic prophylaxis in 2007 [13], the proportion of classis SBP has gradually declined to between 50% and 59% [14,15], which is nearly on par with CNNA [3,16,17]. Between 33.3–58% of patients with suspected ascitic fluid infection have CNNA [18,19], while the prevalence of MNB is between 11% and 26% [19,20].

The 2013 American Association for the Study of Liver Diseases (AASLD) guidelines on the management of adult patients with ascites due to cirrhosis suggests antibiotic therapy

for patients with an AF PMN count of $\geq$250 cells/mm$^3$ or <250 cells/mm$^3$ but with signs of infection [8]. This recommendation implies that all three types of SBP warrant immediate treatment once symptoms become known.

## 4. Pathogenesis

Mechanisms that may be involved in the pathogenesis of SBP are shown in Figure 1.

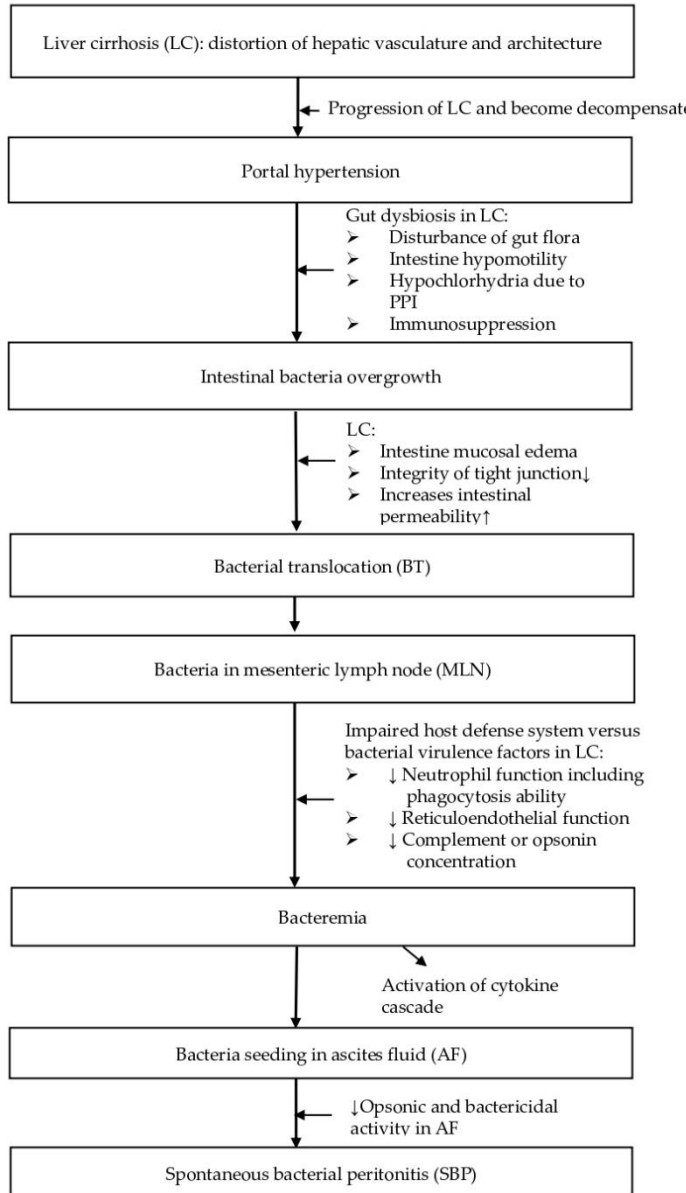

**Figure 1.** Mechanisms involved in the pathogenesis of SBP.

### 4.1. Gut Dysbiosis

One of the early stages in the development of SBP is the disturbance of gut flora leading to bacterial overgrowth and extra-intestinal dissemination of gut microorganisms [21]. Edema of the small intestine and ascending colon alters tight junction integrity and increases intestinal permeability, thus predisposing the patient to bacterial overgrowth in the presence of cirrhosis [21]. Altered small intestinal motility, presence of hypochlorhydria due to the use of proton pump inhibitors, and immunosuppression therapies commonly used during cirrhosis may also contribute to bacterial overgrowth.

*4.2. Bacterial Translocation*

Another important step following bacterial overgrowth is the translocation of enteric bacteria to extraintestinal sites, such as the mesenteric lymph nodes (MLN), which drain lymph from the gut [21]. Bacterial translocation (BT) has been defined as culture-positive MLN [22] and is culture-negative in healthy experimental animals without cirrhosis [22] but culture pathogenic bacteria in 78.1% of animals with cirrhosis and ascites [21]. The fact that SBP is monomicrobial implies that there are "filters" between polymicrobial intestinal sources and the ascitic fluid [21]. The first filter is the gut mucosa itself, and the second filter is the MLN [21]. If these MLN fail to sequester and destroy the bacteria, the pathogens can move from the mesenteric lymphatic system to systemic circulation and then percolate through the liver and extravasate across Glisson's capsule to enter the ascitic fluid [23].

BT has been demonstrated in rats with cirrhosis [21,24] as well as in humans [25]. In a large-scale case study, MLNs were obtained from 101 patients with cirrhosis as well as from 35 non-cirrhotic control participants [25]. Enteric organisms were grown from culture in only 8.6 percent of controls compared to 3.4, 8.1, and 30.8% of patients with Child class A, B, and C cirrhosis, respectively. Selective intestinal decontamination (SID) reduced the rate of positive cultures to that of non-cirrhotic patients.

*4.3. Impaired Host Defense System*

Conversely, the host defense system also plays an important role in SBP pathogenesis. Once a microorganism enters the ascitic fluid, a battle ensues between the invading bacteria and the host's immune system. Peritoneal macrophages are the first line of defense in the peritoneal cavity [26,27]. If these phagocytes fail to eradicate the invading microorganism, the complement system is activated, and cytokines are released [28]. Polymorphonuclear leukocytes (PMNs) then enter the peritoneum to eliminate the foreign bacteria.

However, cirrhotic patients are known to have impairment in neutrophil and reticuloendothelial function [29,30]. In addition, many cirrhotic patients with ascites have a complement or opsonin deficiency [31]. The opsonic activity of ascitic fluid parallels the ascitic fluid total protein concentration [32]. Since opsonins are required by phagocytic cells to eliminate the offending microorganisms, cirrhotic patients with an ascitic fluid protein concentration of less than or equal to ($\leqq$) 1 g/dL are 10-times more likely to develop SBP during hospitalization than those with a protein concentration greater than (>) 1 g/dL [31]. Moreover, each of these abnormalities can individually predispose cirrhotic patients to prolonged bacteremia and peritoneal infection.

## 5. Bacteriology

Bacterial translocation (BT) from the GI tract is the most common source of SBP. However, especially in nosocomial SBP, other sources, such as transient bacteremia due to invasive procedures, can also lead to SBP [33]. The bacteriology of SBP can be classified into Gram-negative bacilli, Gram-positive cocci, multidrug-resistant microorganisms, and anaerobes by bacterial spectrum, and classified into community acquired (CA), healthcare-associated (HCA), and nosocomial (hospital-acquired) infections by facilities.

*5.1. Gram-Negative Bacilli*

Two thirds (66.7%) of SBP cases are caused by Gram-negative bacilli (GNB) from BT, almost exclusively Enterobacteriaceae, and occur independently from the site of acquisition [3,19]. *Escherichia coli* (*E. coli*) is the most frequently isolated pathogen (46–70%) [3,34], followed by *Streptococcus pneumoniae* (18–19%) and *Klebsiella* (9–13%) [19,34]. These three species accounted for approximately 80% to 85% of cases prior to 2007 [11,34].

### 5.2. Gram-Positive Cocci

On the other hand, Gram-positive organisms accounted for less than one third (33.3%) of SBP and were predominated by *Streptococcus* (60%) and *Staphylococcus aureus* (40%) [19,35].

However, a trend of Gram-positive cocci (GPC)-associated SBP has been demonstrated in recent years, representing a changing paradigm in the known bacteriology of SBP [11]. An observational study performed in France demonstrated that GPC-associated SBP was the predominate group, representing 56.1% (coagulase-negative *Staphylococcus* 22.8%, *Enterococcus* 22.8%) of SBP cases [36]. The phenomenon has been linked to invasive therapeutic intervention [3] as well as long-term quinolone administration [37–39]. For instance, a retrospective study performed in Spain found a significant difference in the frequency of SBP caused by GPC among patients not provided (30.2%) and provided (78.6%) norfloxacin prophylaxis [38].

### 5.3. Multidrug-Resistant Microorganisms

The emergence of multidrug-resistant (MDR) microorganisms, such as extended-spectrum β-lactamase (ESBL)-producing GNB and enterococci [40], fluroquinolone-resistant (QR) GNB [41], cefoxitin/methicillin-resistant Staphylococcus aureus (MRSA) [11], vancomycin-resistant enterococcus (VRE), and other resistant microorganisms [40] have altered prior conceptions toward SBP bacteriology and treatment [42]. One study performed in Greece demonstrated that 20.8% of isolated bacteria in patients with culture-positive SBP and spontaneous bacteremia (SB) were MDR, including ESBL-GNB, *P. aeruginosa* and *E. faecium* [43]. Furthermore, 10% of the isolated strains were extensively drug resistant (XDR) [43]. A prospective study performed in Spain found that ESBL Enterobacteriaceae (*E. coli* and *Klebsiella*) were the most common multi-drug resistant bacteria (73%), especially among nosocomial infections, followed by QR GNB in patients with long-term norfloxacin prophylaxis [41]. Meanwhile, MRSA was isolated in 9.2% cases of SBP during another prospective study in France [44]. In another study launched in Greece, SBP due to GPC was found in 55% of cases when patients received quinolone prophylaxis, and MRSA was the most common isolate (8.5%) [40]. Regarding nosocomial infections, the prevalence of cefoxitin/methicillin-resistant Staphylococcus aureus (MRSA) was 24.8% in a study launched in France [45]. In comparison with non-MRSA infections, MRSA infections were more likely to recur and occurred in more sites other than ascitic fluid and blood ($p < 0.0004$) [45].

### 5.4. Anaerobes

Although gut floras are responsible for the majority of SBP cases, anaerobes appear to be rare, presumably due to the high oxygen content of the intestinal wall and AF, as well as because of the relative inability of anaerobes to translocate across the intestinal mucosa [46,47].

### 5.5. Community-Acquired SBP vs. Healthcare-Associated SBP vs. Nosocomial-Acquired SBP

A study performed in Spain showed that Enterobacteriaceae (*E. coli*, K. pneumoniae) from BT is the leading cause of infection in CA SBP, while GPC from invasive procedures and ICU treatment is the primary cause of infection in nosocomial SBP [41]. A study performed in China also demonstrated more frequent GPC (especially Enterococcus) in the nosocomial group when compared with the CA group (16.6% vs. 9.0%, $p < 0.05$) [48]. The other study performed in France confirmed the above result and revealed that nosocomial and staphylococcal infections were associated with a higher mortality rate than community-acquired infections ($p = 0.0255$) and non-staphylococcal infections ($p < 0.001$), respectively [45]. In addition, a high prevalence of multi-drug resistant bacteria was found in nosocomial SBP (22% versus 2% in CA SBP versus 5% in HCA), with extended-spectrum β-lactamase-producing Enterobacteriaceae (ESBL-E) being the main multi-resistant organism identified [41].

## 6. Diagnosis

### 6.1. Clinical Presentations

Spontaneous bacterial peritonitis (SBP) should be suspected in patients with cirrhosis who develop signs or symptoms, such as fever (69%), abdominal pain (59%), altered mental status (54%), abdominal tenderness (49%), diarrhea (32%), ileus (30%), hypotension/shock (21%), or hypothermia (17%) [46]. However, 10% of cases show no signs or symptoms, partly because a large volume of ascites prevents contact of the visceral and parietal peritoneal surfaces to elicit the spinal reflux that cause abdominal rigidity [46].

### 6.2. Diagnostic Paracentesis

A diagnostic paracentesis should be performed in all patients with cirrhosis and ascites who require emergency room care or hospitalization, who demonstrate or report signs/symptoms mentioned above in the clinical presentations, or who present gastrointestinal bleeding, in order to confirm evidence of SBP [49]. However, low clinical suspicion for SBP does not preclude the necessity for paracentesis, since 10% of cases have no signs or symptoms [46]. In fact, AF infection is the most frequent complication among patients with cirrhosis and ascites, accounting for 31% of all bacterial infections [46]. Patients who underwent paracentesis had a lower in-hospital mortality rate than those who did not (6.5% versus 8.5%; adjusted odds ratio 0.55, 95% CI 0.41–0.74) in a study of 17,711 patients with cirrhosis and ascites [50]. Paracentesis should be avoided only in instances of clinically evident fibrinolysis or disseminated intravascular coagulation [46].

Notably, delayed paracentesis could lead to a 2.7-fold increased risk of in-hospital mortality in patients with SBP after adjusting for MELD score and renal dysfunction [51]. One retrospective study performed in the U.S. revealed that diagnostic paracentesis performed <12 h after hospitalization in patients with cirrhosis and ascites may improve short-term survival [51].

Paracentesis has been shown to be safe despite the expected coagulopathy in these patients. There is an approximate 1% chance of significant abdominal-wall hematoma, 0.01% chance of hemoperitoneum, and 0.01% chance of iatrogenic infection associated with paracentesis [46]. Still, paracentesis should be performed by well-trained personnel who have completed 3–10 paracentesis under supervision by an experienced clinician [52]. Caution is advised when performing paracentesis, especially in patients with ileus, multiple surgical scars, risk of severe bleeding, or when the clinician lacks experience [11]. In such cases, ultrasound guidance may be helpful. Routine correction of prolonged prothrombin time or thrombocytopenia is not required when experienced personnel perform the paracentesis [52].

### 6.3. Handling and Interpretation of Ascites Fluid Study

Ascitic fluid tests should include cell count with a differential, Gram stain, culture, total protein, and albumin to calculate the serum-ascites albumin gradient (SAAG), if not already known [11]. When the diagnosis of cirrhosis is not definite, an ascites SAAG greater than or equal to (≧) 1.1 g/dl is ascribed to portal hypertension with approximately 97% accuracy [49]. Total ascitic fluid protein concentration should be measured to assess the risk of SBP since patients suffering from ascites with a total protein concentration lower than (<) 1.5 g/dL are at increased risk of SBP [7].

Regarding cell count, approximately 1 mL of fluid should be injected into a purple-top ethylenediaminetetraacetic acid (EDTA) tube, which contains anti-coagulant to avoid clotting and inaccurate interpretation [11].

For bacterial culture, AF should be inoculated into blood culture bottles at the bedside [18]. At least 10 mL of AF in both aerobic and anaerobic blood culture bottles can increase the percentage of cases with positive cultures and accelerate the detection time for bacterial growth [53]. However, up to 60% of ascites culture may be negative in patients with clinical manifestations, suggestive of SBP and increased ascites neutrophil count [54].

### 6.4. Diagnostic Criteria for SBP Types, Distinguishing Secondary Bacterial Peritonitis

A diagnosis of (1) classic SBP is made if PMN count in the ascitic fluid is $\geq$250 cells/mm$^3$, culture results are positive, and secondary causes of peritonitis are excluded [7,49]. A potential source of error in PMN count is hemorrhage into the ascitic fluid, such as with traumatic paracentesis, which can cause both red and white blood cells to enter the ascites. A corrected PMN count should be calculated if there are bloody ascites by subtracting one PMN from the absolute PMN count for every 250 red cells/mm$^3$ [55].

There are two other types of SBP, (2) CNNA (culture-negative neutrocytic ascites) and (3) MNB (monomicrobial nonneutrocytic bacterascites). (2) CNNA is diagnosed when AF culture is negative, PMN cell count is <250/mm$^3$, and there is no surgically treatable intra-abdominal source of infection [56]. (3) MNB is characterized by a positive ascites culture and PMN cell count < 250/mm$^3$ (Table 1).

Distinguishing SBP from secondary bacterial peritonitis is essential because the conditions require different therapeutic strategies. Mortality from SBP can be as high as 85% if a patient undergoes an unnecessary exploratory laparotomy [57], while mortality of secondary bacterial peritonitis can exceed 80% if treatment consists of antibiotics without surgical intervention [9]. AF in secondary peritonitis usually meets at least two of the following criteria (sometimes referred to as Runyon's criteria): ① Ascites total protein content is >1 g/dL, ② an ascites glucose concentration of <50 mg/dL, and ③ an ascites lactate dehydrogenase level of >225 U/mL (or higher than the upper limit of normal in serum) [9]. In addition, a Gram stain demonstrating a number of bacterial forms or cultures showing a polymicrobial infection implies gut perforation [58]. At least two of Runyon's criteria were met and/or the presence of a polymicrobial ascitic fluid culture are present in 96% of patients with secondary bacterial peritonitis [59].

### 6.5. Other Diagnostic Markers of SBP

Several studies showed that either ascites calprotectin or the ratio of calprotectin to total protein has diagnostic and prognostic value for SBP in patients with liver cirrhosis and ascites [60,61]. Calprotectin is a calcium- and zinc-binding protein detected almost exclusively in neutrophils, and its presence in body fluids is proportional to the influx of neutrophils [4]. AF calprotectin is related PMNL. This suggests that it may be a suitable candidate for diagnosis of SBP. In addition, the presence of fecal calprotectin quantitatively relates to intestinal neutrophil migration and is therefore considered as a marker of intestinal inflammation and hence a screening tool for SBP [62].

## 7. Risk Factors

### 7.1. First Episode

Since SBP may be regarded as the final clinical stage of liver cirrhosis [63], liver disease severity is the most important risk factor. One Spanish study found that serum bilirubin concentration of >2.5 mg/dL and AF total protein of <1.0 g/dL were independent risk factors for experiencing an initial SBP episode [64]. Another Spanish study also showed that AF total protein of <1.5 g/dL combined with advanced liver failure (serum bilirubin level of $\geq$3 mg/dL with a Child–Pugh score of $\geq$9 points) or with impaired renal function (serum creatinine level of >1.2 mg/dL, blood urea nitrogen level of >25 mg/dL, or serum sodium level of <130 mEq/L) is associated with an increased risk of SBP [13]. Since a significant correlation between AF opsonic activity and AF protein concentration has been observed [65], a simple measurement of AF protein could replace AF opsonic activity as a predictor of the occurrence of SBP [64].

Other common risk factors for initial SBP onset include acute variceal bleeding (7.9% incidence of SBP even under antibiotic prophylaxis) [66–68], acid suppressive therapy especially proton pump inhibitor (PPI) as evident by three major meta-analyses [69–71], and asymptomatic bacteriuria [72].

In contrast, nonselective beta blockers (NSBB, propranolol) were found to be protective against initial SBP onset, as reported in a meta-analysis reviewing three randomized

controlled trials and three retrospective studies, which demonstrated a statistically significant difference (12.1%, $p < 0.001$) in favor of taking propranolol than no treatment in preventing SBP in ascitic cirrhosis [73]. A genetic variability in the tumor necrosis factor-$\alpha$ (TNFA) c.-238A allele was associated with a decreased risk of severe bacterial infections in patients with end-stage liver disease awaiting liver transplantation (hazard ratio 0.43, 95% CI 0.20–0.91) [74].

### 7.2. Recurrent Episodes

Prior to antibiotic prophylaxis, patients who survived an initial SBP episode were at elevated risk of recurrence: 43% by 6 months, 69% by 1 year, and 74% by 2 years [75]. Risk factors include serum albumin level lower than 2.85 g/dL at hospital discharge [4], low ascitic fluid protein concentration (<1g/dL), low prothrombin (≤45%) [75], and a prior episode of SBP.

## 8. Prognosis

### 8.1. In-Hospital Mortality

When first reported, the in-hospital mortality of an episode of SBP exceeded 90%; however, the rate has been reduced to approximately 20% through early diagnosis and prompt antibiotic therapy [6,76].

Regarding prognostic factors for SBP, a meta-analysis selecting the 12 best-quality studies revealed that renal dysfunction is the main prognostic factor for cirrhotic patients with SBP (mortality rates of 67% versus 11% in those with and without renal dysfunction, respectively), followed by the MELD score [77]. Nosocomial (49.5%) and Staphylococcal infections (65.3%) were associated with a higher mortality rate than community-acquired infections (23.8%) ($p = 0.0255$) and non-Staphylococcal infections ($p < 0.001$), respectively [45]. Although overall mortality among patients with SBP-associated septic shock was 81.8%, a retrospective study demonstrated that timely (<3 h) and appropriate antimicrobial therapy could significantly decrease mortality (OR = 0.54, $p = 0.02$) [78].

### 8.2. Long-Term Mortality

Since SBP may be regarded as the final clinical stage of liver cirrhosis [63], one-year overall mortality rates range from 53.9 [76] to 78% [64,76,79]. In a large database study of 16,922 patients with cirrhosis, the one, two, and three-year mortality rate for patients following hospitalization due to SBP was 53.9%, 61.4%, and 66.5%, respectively [76]. Thus, liver transplantation should be seriously considered for SBP survivors who are good candidates for transplantation [46].

## 9. Treatment

The standard treatment for SBP is prompt broad-spectrum antibiotic administration and albumin supplementation, especially in patients with renal impairment (RI) [54].

### 9.1. Antibiotic Therapy

If SBP is suspected, antibiotic therapy must be initiated immediately after AF and culture to reduce complications and mortality [7]. Potentially nephrotoxic antibiotics (i.e., aminoglycosides) should be avoided [80] since patients with SBP are highly sensitive to aminoglycosides-related nephrotoxicity and fatal renal failure is common even at sub-toxic doses [46].

Two decades ago, most cases of SBP were attributed to third generation cephalosporin-sensitive Enterobacteriaceae. Now, risk factors, such as repeated hospitalizations, invasive procedures, and frequent exposure to antibiotics either as prophylaxis or as treatment [3], have led to the development of infections caused by MDR microorganisms. Bacterial resistance carries a 3.87-fold relative increased risk of mortality in patients with SBP [33]. Particularly, nosocomial SBP has been associated with multi-drug resistance (HR = 4.43) and poor outcome (50% in-hospital mortality). One prospective study demonstrated that

failure of recommended empirical antibiotic regimens can have a negative impact on mortality [81]. Therefore, it is important to distinguish community-acquired SBP from healthcare-associated and nosocomial SBP, as well as consider the severity of infection and local resistance profile before implementing antibiotic therapy [54] (Figure 2). Subsequently, de-escalation according to bacterial susceptibility based on positive culture is recommended to minimize resistance selection pressure.

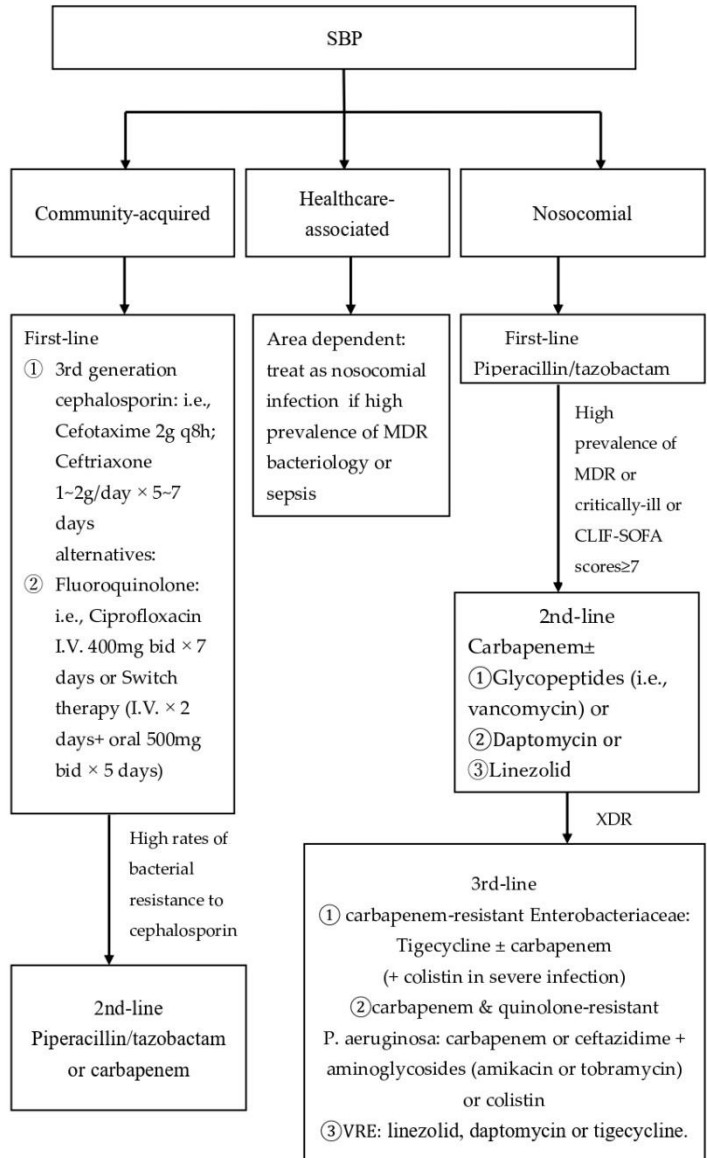

**Figure 2.** Recommended antibiotic treatment strategy for SBP since 2018.

### 9.1.1. Community-Acquired SBP

Third generation cephalosporin is a reasonable choice for suspected community-acquired SBP [54,82]. Cefotaxime was extensively studied in patients with SBP in the 1980–90s due to its coverage of most causative organisms and excellent ascitic fluid concentrations [7,83]. Infection resolution was achieved in 77% to 98% of patients [54]. A dose of 2 g every 12 h was as effective as 2 g every 6 h in one randomized multicenter study [84]. In addition, a five-day course of this drug is as effective as a 10-day course [85]. Clinically, cefotaxime is given at a dose of 2 g intravenously every eight hours and is also effective in treating bacteremia [46]. Alternative regimens include ceftriaxone 1 to 2 g/day for 5–7 days [86–88] and ciprofloxacin 400 mg twice daily for 7 days in patients with normal renal function [89]. Ciprofloxacin can be used in patients who cannot take cephalosporin,

although this drug does not penetrate to the ascitic fluid as well as cefotaxime. Since ciprofloxacin is more expensive than cephalosporin, a cost-effective ciprofloxacin replacement therapy can be substituted by first administering ciprofloxacin 200 mg intravenously twice daily for two days, then changing to oral ciprofloxacin 500 mg or ofloxacin 400 mg twice daily for a total 7-day course [14,89,90]. Of note, fluoroquinolones should not be used in patients who had previously received fluoroquinolone for SBP prophylaxis because the offending microorganism may be resistant to fluoroquinolones. Fortunately, microorganisms that infect patients who have been on fluoroquinolone prophylaxis are usually responsive to cefotaxime [38].

Collectively, third generation cephalosporins are recommended as first-line antibiotic treatment for community-acquired SBP in countries with low rates of bacterial resistance. However, in countries with high rates of bacterial resistance, piperacillin/tazobactam or carbapenem should be considered [54] (see below).

### 9.1.2. Nosocomial and Healthcare SBP

Since bacterial resistance to third generation cephalosporins is higher in nosocomial (40.9%) and healthcare-associated SBP (21.1%) versus community-acquired cases (7.1%) [91], piperacillin/tazobactam have been recommended as a first-line therapy for nosocomial and healthcare-associated SBP in areas with low prevalence of infection by MDR bacteria [54].

Conversely, carbapenem alone or in combination with glycopeptides, with daptomycin [92], or with linezolid has been recommended as a first-line therapy for nosocomial and healthcare SBP with high severity (critically ill or CLIF-SOFA scores of $\geq 7$) [93] or in areas with a high prevalence of Gram-positive MDR bacteria, such as MRSA [54,94]. Currently, vancomycin, the chief antibiotic for infection by MRSA for years, has limitations in terms of in vitro sensitivity, toxicity, and overall patient management. Due to the increase in the mean MICs of vancomycin in clinical isolates of S. aureus, daptomycin could be considered a priority due to its less hepatic metabolism, renal toxicities and myelotoxicity [95]. In addition, many novel antibiotics for MDR Gram-positive microorganisms are being developed, but only with appropriate dosing, utilization and careful monitoring for the emergence of antimicrobial resistance can ensure these current antimicrobial drugs continue to treat Gram-positive pathogens in the future [96].

XDR, such as carbapenem-resistant Enterobacteriaceae, can be treated with tigecycline alone or in combination with a carbapenem via continuous infusion. Addition of intravenous colistin may be necessary for severe infections [54]. *Pseudomonas aeruginosa* resistant to carbapenems and quinolones usually require carbapenem or ceftazidime in combination with aminoglycosides (amikacin or tobramycin) or colistin. Vancomycin resistant Enterococci (VRE)-related SBP should be treated with linezolid, daptomycin, or tigecycline.

When using highly nephrotoxic antibiotics, such as vancomycin or aminoglycosides, serum levels must be monitored closely to decrease the risk of renal failure.

### 9.2. Indication for Repeated Paracentesis

Most cases of SBP respond well to a five-day antibiotic course, except in patients whose culture show growth of unusual microorganisms (e.g., pseudomonas), microorganisms resistant to standard antibiotic therapy, or microorganisms associated with endocarditis (e.g., *Staphylococcus aureus* or *Streptococcus viridians*) that may require longer treatment duration [85]. After the scheduled course, patients should be reassessed. Treatment should be discontinued in case of unanticipated, rapid improvement. However, if fever or pain persists, paracentesis should be repeated. If ascites PMN count is elevated but less than pre-treatment value, antibiotics should be continued for another 48 h and paracentesis repeated [46]. However, if ascites PMN count is greater than pre-treatment value, a search for secondary bacterial peritonitis should be performed urgently.

*9.3. Albumin Supplement in Patients with Renal Impairment*

The administration of albumin (given 1.5 g/kg at diagnosis and 1 g/kg on day 3, maximum 100 g) is recommended in patients with SBP [97], especially in those with serum creatinine of >1 mg/dL, blood urea nitrogen of >30 mg/dL, or total bilirubin of >4 mg/dL [98]. In one meta-analysis of four RCTs (288 patients), albumin infusion prevented renal impairment and reduced mortality among patients with SBP [99].

*9.4. Discontinue NSBB in Patients with SBP*

Given that NSBB may interfere with systemic hemodynamics, effect on outcome was examined in a retrospective study of 607 patients with cirrhosis and ascites [100]. Once SBP developed, patients receiving NSBB suffered lower transplant-free survival (HR = 1.58; 95% CI: 1.098–2.274). In addition, a higher proportion of patients on NSBBs suffered hepatorenal syndrome (24% vs. 11%, $p$ = 0.027) [100]. Therefore, NSBB should be discontinued after a diagnosis of SBP.

*9.5. Other Novel Therapeutic Strategies*

The emergence of MDR bacteria may require novel therapeutic strategies that do not involve the use of antibiotics. One potential approach is human amniotic mesenchymal stromal cell (hA-MSC) treatment. One in vitro study found hA-MSCs added to ascites fluid could significantly reduce the proliferation of both bacterial strains at 24 h as well as affect M1/M2 polarization, C3a complement protein, and ficolin 3 concentrations during the course of infection in a strain-dependent manner [101]. Validation of an in vivo model is warranted for future hA-MSC application in treating ascites infected with carbapenem-resistant bacteria [101].

## 10. Prophylaxis of SBP

*10.1. SBP Prophylaxis in High-Risk Groups*

Not all patients with cirrhosis and ascites require antibiotic prophylaxis, sometimes referred to as selective intestinal decontamination (SID). SID is associated with a reduced risk of bacterial infection [37,102–104] and mortality [13,105,106]. However, the use of long-term SID can accelerate the selection of resistant bacteria that may subsequently cause spontaneous infection [107,108]. Antibiotic resistance to third generation cephalosporins or quinolones could lead to poor prognosis and antibiotic failure [109]. In addition, SID improves 3-month and 1-year survival, but data on longer SID courses are not available [13]. Therefore, SID is only recommended for patients at high-risk of SBP to prevent the occurrence or recurrence of SBP.

Three high-risk patient groups have been identified that may benefit from SID: (i) patients with acute upper gastrointestinal bleeding (UGIB); (ii) patients with low ascites total protein content, associated advanced liver failure or renal dysfunction, and absence of prior SBP (primary prophylaxis), and (iii) patients with a history of SBP (secondary prophylaxis). At least three meta-analyses have supported the benefit of SID in these high-risk group [105,110,111]. Notably, a recent meta-analysis argued that there is considerable uncertainty about whether antibiotic prophylaxis is beneficial, as well as which antibiotic type is beneficial to patient with cirrhosis and ascites [112]. More randomized clinical trials with adequate power are needed [112].

10.1.1. Prophylaxis in Patients with Upper Gastrointestinal Bleeding (UGIB)

UGIB increases the risk of SBP and other infections during or after a bleeding episode (first 7 days), with an incidence between 16% (compensated cirrhosis) and 66% (advanced cirrhosis) [113,114]. Infections synergistically increase the probability of uncontrollable bleeding, recurrent bleeding [115], and hospital mortality [105].

Conversely, SID reduces the risk of mortality, infections associated with SBP, and repeat bleeding in patients who are hospitalized with cirrhosis and UGIB. In one meta-analysis of 12 trials involving a total of 1241 patients with cirrhosis and GI bleeding, antibiotic

prophylaxis was compared to placebo and no antibiotic prophylaxis. The benefit of SID was demonstrated with regard to mortality (relative risk [RR] 0.79, 95% CI 0.63–0.98), bacterial infection (RR 0.35, 95% CI 0.26–0.47), and repeat bleeding (RR 0.53, 95% CI 0.38–0.74) [116]. Therefore, SID should be instituted as early as possible after UGIB according to the recent Baveno VII consensus [117], the American Association for the Study of Liver Diseases [118], and the European Association for the Study of the Liver guidelines [54].

Intravenous ceftriaxone 1 g/24 h is the clinical antibiotic of choice given the high prevalence of quinolone-resistant bacterial infections and conforms to local resistance epidemiology and antimicrobial policies [117]. The antibiotic duration should be a maximum of 7 days (consider discontinuing when hemorrhage has resolved or when vasoactive drugs are discontinued). For patients discharged before seven days of intravenous antibiotic therapy are complete, it is recommended that physicians transition to an oral antibiotic, such as ciprofloxacin (500 mg every 12 h) to complete the seven-day course [113].

### 10.1.2. Primary Prophylaxis in Patients with Low Ascites Total Protein and Associated Advanced Liver Failure or Renal Dysfunction without History of SBP

As mentioned in the "Risk Factors" section, low ascites total protein concentration (<1–1.5 g/dL) increases the risk of an initial SBP episode. However, in the absence of additional risk factors, incidence of SBP is relatively low (<20% at 1 year) [113]. Fernandez, et al. randomized 68 patients with cirrhosis and low ascites protein levels (<1.5 g/dL) with advanced liver failure (Child–Pugh score $\geq$ 9 points with serum bilirubin level $\geq$ 3 mg/dl) or impaired renal function (serum creatinine level of $\geq$1.2 mg/dl, blood urea nitrogen level of $\geq$25 mg/dl, or serum sodium level of $\leq$130 mEq/L) to receive norfloxacin (400 mg/day for 12 months) or placebo to study primary prophylaxis for SBP [13]. Norfloxacin significantly improved three-month survival (94% vs. 62%; $p$ = 0.03), although the significance in survival was lost at one year (60% vs. 48%; $p$ = 0.05). Norfloxacin administration significantly reduced the one-year probability of developing SBP (7% vs. 61%, $p$ < 0.001) and hepatorenal syndrome (28% vs. 41%, $p$ = 0.02), and improved survival at three months (94% vs. 62%, $p$ = 0.003). Another double-blind placebo-controlled trial involving 100 patients with ascitic fluid total protein levels of <1.5 g/dL reported improved survival at one year among patients receiving ciprofloxacin (500 mg/day for 12 months) versus placebo (86% vs. 66%; $p$ < 0.04). Three meta-analyses support a significant preventive effect against SBP (RR 0.18; 95% CI 0.09–0.35; $p$ = 0.001) [105,110,111] despite improvement to survival peaking at three months, decreasing at six months, and dropping off after 12 months of follow-up [54].

Norfloxacin is a poorly absorbed quinolone that selectively inhibits GNB without affecting the anaerobic population since anaerobes are required to maintain the stability of the intestinal flora as well as prevent overgrowth of other offending organisms. SID by norfloxacin 400 mg per day or ciprofloxacin 500 mg per day, or alternatively, cotrimoxazole (160 mg trimethoprim and 800 mg sulfamethoxazole) per day [5] during hospitalization has proven useful in reducing the incidence of SBP as well as the incidence of extraperitoneal infections and short-term mortality.

### 10.1.3. Secondary Prophylaxis in Patients with Prior SBP

Among SBP survivors, the cumulative recurrence rate at one year is between 43 and 70% [4,7]. One randomized, double-blind, placebo-controlled trial of norfloxacin (400 mg/day orally) in patients with a history of SBP found the chance of SBP recurrence fall from 68% to 20% [37]. Another open-label, randomized study compared norfloxacin 400 mg/day to rufloxacin 400 mg/week in the prevention of SBP recurrence [119]. Although the one-year probability of SBP recurrence was not significantly different to control (26% versus 36%, $p$ = 0.16), norfloxacin was effective in the prevention of SBP recurrence due to Enterobacteriaceae (0% vs. 22%, $p$ = 0.01) [119].

Alternatives to norfloxacin 400 mg/day include ciprofloxacin 500 mg once a day or cotrimoxazole (160 mg trimethoprim and 800 mg sulfamethoxazole) per day [5,54].

One meta-analysis found that rifaximin may be effective for both primary and secondary SBP prophylaxis compared to both no antibiotic intervention and systemically absorbed antibiotic administration [120]. However, additional prospective studies are required before a change in clinical practice can be recommended [5]. The use of intermittent ciprofloxacin has been associated with a higher rate of quinolone-resistant infection and should be avoided [121].

## 11. Conclusions

Spontaneous bacterial peritonitis (SBP) is a severe complication in cirrhosis patients with ascites. Clinical awareness, prompt diagnosis by exclusion of secondary bacterial peritonitis, and immediate treatment are necessary to reduce mortality and morbidity in this patient group. However, the emergence of multidrug-resistant (MDR) microorganisms have changed our understanding of SBP bacteriology and treatment. Antibiotic therapy specific to either community-acquired or nosocomial/healthcare-acquired SBP is ideal, while liver transplantation remains the definitive treatment following SBP. Prevention of SBP recurrence by antibiotic prophylaxis while patients wait for a liver transplant is therefore an important clinical issue. The poorly absorbed antibiotic rifaximin may be effective for both primary and secondary SBP prophylaxis, but additional prospective studies are required. Further development of non-antibiotic strategies based on pathogenic mechanisms are also urgently needed. Blind studies that avoid post-randomization dropout and consider clinically relevant outcomes, such as mortality, health-related quality of life, and decompensation events, are desired for future research.

There are three types of SBP. Bacterial translocation from the GI tract is the most common source of SBP. Therefore, two thirds of SBP cases were caused by Gram-negative bacilli, almost exclusively Enterobacteriaceae. *Escherichia coli* (*E. coli*) is the most frequently isolated pathogen. However, a trend of Gram-positive cocci (GPC)-associated SBP has been demonstrated in recent years, representing a changing paradigm in the known bacteriology of SBP, especially in nosocomial SBP; other sources, such as transient bacteremia due to invasive procedures, can also lead to SBP Gram-positive cocci (GPC), such as Staphylococcus, Enterococcus, as well as multi-resistant bacteria have become common pathogens and have changed the conventional approach to treatment of SBP. Healthcare-associated and nosocomial SBP infections should prompt greater vigilance and consideration for alternative antibiotic coverage. Acid suppressive and beta-adrenergic antagonist therapies are strongly associated with SBP in at-risk individuals. A diagnostic paracentesis should be performed in all patients with cirrhosis and ascites who require emergency room care or hospitalization, who demonstrate or report signs/symptoms mentioned above in the clinical presentations, or who present gastrointestinal bleeding, in order to confirm evidence of SBP. Distinguishing SBP from secondary bacterial peritonitis is essential because the conditions require different therapeutic strategies. Since SBP may be regarded as the final clinical stage of liver cirrhosis [63], one-year overall mortality rates range from 53.9 [76] to 78% [64,76,79]. In a large database study of 16,922 patients with cirrhosis, the one, two, and three-year mortality rate for patients following hospitalization due to SBP was 53.9%, 61.4%, and 66.5%, respectively [76]. Thus, liver transplantation should be seriously considered for SBP survivors who are good candidates for transplantation. The standard treatment for SBP is prompt broad-spectrum antibiotic administration and should be tailored according to either CAP or hospital-acquired, or to local resistance profiles. Albumin supplementation, especially in patients with renal impairment (RI) is also beneficial. Not all patients with cirrhosis and ascites require antibiotic prophylaxis, sometimes referred to as selective intestinal decontamination (SID). SID is associated with a reduced risk of bacterial infection [37,102–104] and mortality.

**Author Contributions:** Conceptualization, review, editing, supervision, and funding: C.-H.H.; manuscript drafting: C.-H.H., C.-H.L. and C.C. English editing: C.-H.L. All authors have read and agreed to the published version of the manuscript.

**Funding:** This study was supported by grants from National Science and Technology Council, Taiwan (MOST 110-2314-B-182A-093), and from Chang Gung Medical Research Project (CMRPG3K0401-0402, NMRPG3L0331).

**Institutional Review Board Statement:** Ethical review and approval were waived for this study given that it is a literature review.

**Informed Consent Statement:** Patient consent was waived given that it is a literature review.

**Data Availability Statement:** Not applicable.

**Acknowledgments:** We thank Chang Gung Memorial Hospital and Ministry of Science and Technology, Taiwan for their grant support.

**Conflicts of Interest:** The authors declare no conflict of interest.

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
