# Peer review of "Spontaneous Bacterial Peritonitis in Decompensated Liver Cirrhosis—A Literature Review"

_livers, doi:10.3390/livers2030018_

Round 1
Reviewer 1 Report
Dear author/
its of my pleasure to read this valuable review. It need major revision then re submit it.

Author Response
Reviewer Comment:
Comment: The language quality of your manuscript is not sufficient:
Response: The manuscript has been edited in English
Reviewer: 1
Comment to the Author
Comment 1: More details regarding the source of SBP; hospital or community acquired
Response 1: Thanks for the good comment. We have added the above information on page 4 to 5 in the” Bacteriology” section in red marks and as follows:
“ Bacterial translocation (BT) from GI tract is the most common source of SBP. However, especially in nosocomial SBP, other sources such as transient bacteremia due to invasive procedures can also lead to SBP(1). The bacteriology of SBP can be classified into gram-negative bacilli, gram-positive cocci, multidrug-resistant microorganisms, and anaerobes by bacterial spectrum, and classified into community acquired (CA), healthcare associated (HCA), and nosocomial(hospital-acquired) infections by facilities.
A study performed in Spain showed that Enterobacteriaceae (E. coli, K. pneumoniae) from BT is the leading cause of infection in CA SBP, while GPC from invasive procedure and ICU treatment is the primary cause of infection in nosocomial SBP (2). A study performed in China also demonstrated more frequent GPC(especially Enterococcus) in the nosocomial group when compared with the CA group (16.6% vs. 9.0%, P < 0.05) (3).The other study performed in France confirmed the above result and revealed that nosocomial and staphylococcal infections were associated with a higher mortality rate than were community-acquired infections (P=.0255) and non-staphylococcal infections (P<.001), respectively(4). In addition, a high prevalence of multi-drug resistant bacteria was found in nosocomial SBP (22% versus 2% in CA SBP versus 5% in HCA), with extended-spectrum β-lactamase-producing Enterobacteriaceae (ESBL-E) being the main multi-resistant organism identified(2).
Comment 2: Regarding the bacteriological etiology, more details regarding why gram negative is more prevalent and why E. coli is prevailed in SBP.
Response 2: Thanks for the good comment. We have added this information on page 4 to 5 in the” Bacteriology” section in red marks and as follows:
Bacterial translocation (BT) from GI tract is the most common source of SBP. Two thirds (66.7%) of SBP cases are caused by gram-negative bacilli (GNB) from BT, almost exclusively Enterobacteriaceae, and occur independently from the site of acquisition (5, 6). Escherichia coli (E. coli) is the most frequently isolated pathogen (46-70%) (7, 8), followed by Streptococcus pneumoniae (18-19%) and Klebsiella (9-13%).
Comment 3: Gram positive bacteria need to be mentioned in its section not in gram negative section, also, more details
Response 3: Thanks for the good comment. We have mentioned Gram positive bacteria in 5.2 Gram-positive cocci section and added more information regarding GPC on page 4 to 5 in the” Bacteriology” section in red marks and as follows:
However, especially in nosocomial SBP, other sources such as transient bacteremia due to invasive procedures can also lead to SBP(1)…. A study performed in Spain showed that Enterobacteriaceae (E. coli, K. pneumoniae) from BT is the leading cause of infection in CA SBP, while GPC from invasive procedure and ICU treatment is the primary cause of infection in nosocomial SBP (2). A study performed in China also demonstrated more frequent GPC(especially Enterococcus) in the nosocomial group when compared with the CA group (16.6% vs. 9.0%, P < 0.05) (3).The other study performed in France confirmed the above result and revealed that nosocomial and staphylococcal infections were associated with a higher mortality rate than were community-acquired infections (P=.0255) and non-staphylococcal infections (P<.001), respectively(4)
Comment 4: The abstract is not well organized to reach to conclusions
Response 4: Thanks for the good comment. We have rewritten the abstract on page 1 and as follows:
Background: Spontaneous bacterial peritonitis (SBP) is defined as a bacterial infection of the ascitic fluid without surgically treatable intra-abdominal infection source. SBP is a common, severe complication in cirrhosis patients with ascites, and if left untreated, in-hospital mortality may exceed 90%. However, the incidence of SBP has been lowered to approx. 20% through early diagnosis and antibiotic therapy. Clinical awareness, prompt diagnosis, and immediate treatment are advised when caring for these patients to reduce mortality and morbidity. Aim: To discuss important issues comprising types of SBP, pathogenesis, bacteriology including the emergence of multidrug-resistant (MDR) microorganisms, prompt diagnosis, risk factors, prognosis, treatment strategies, as well as recurrence prevention through antibiotic prophylaxis until liver transplantation and future trends in treating and preventing SBP in detail. Methods: This article is a literatures review and appraisal of guidelines, randomized controlled trials, meta-analyses, and other review articles found on PubMed from between 1977 and 2022. Results: There are three types of SBP. Bacterial translocation from GI tract is the most common source of SBP. Therefore, two thirds of SBP cases were caused by gram-negative bacilli, of which Escherichia coli is the most frequently isolated pathogen. However, a trend of gram-positive cocci associated SBP has been demonstrated in recent years, possibly related to more invasive procedures and long-term quinolone prophylaxis. A diagnostic paracentesis should be performed in all patients with cirrhosis and ascites who require emergency room care or hospitalization, who demonstrate or report consistent signs/symptoms in order to confirm evidence of SBP. Distinguishing SBP from secondary bacterial peritonitis is essential because the conditions require different therapeutic strategies. The standard treatment for SBP is prompt broad-spectrum antibiotic administration and should be tailed according to community-acquired SBP, health care-associated or nosocomial SBP infections and local resistance profile. Albumin supplementation, especially in patients with renal impairment, is also beneficial. Selective intestinal decontamination is associated with a reduced risk of bacterial infection and mortality in high-risk group. Conclusions: The standard treatment for SBP is prompt broad-spectrum antibiotic administration and should be tailed according to community-acquired SBP, health care-associated or nosocomial SBP infections and local resistance profile. Since the one-year overall mortality rates for SBP ranges from 53.9 to 78%, liver transplantation should be seriously considered for SBP survivors who are good candidates for transplantation. Further development of non-antibiotic strategies based on pathogenic mechanisms are also urgently needed.
Comment 5: The introduction needs more organization
Response 5: Thanks for the good comment. We have reorganized the introduction on page 2.
Comment 6: Inapparent key points in the paper make it unique.
Response 5: Thanks for the good comment. We have summarized key points in the abstract and added novel information on page 12 “Other novel therapeutic strategies” section and as follows:
The emergence of MDR bacteria may require novel therapeutic strategies that do not involve the use of antibiotics. One potential approach is human amniotic mesenchymal stromal cell (hA-MSC) treatment. One in vitro study found hA-MSCs added to ascites fluid could significantly reduce the proliferation of both bacterial strains at 24 h as well as affect M1/M2 polarization, C3a complement protein, and ficolin 3 concentrations during the course of infection in a strain-dependent manner (9). Validation of an in vivo model is warranted for future hA-MSC application in treating ascites infected with carbapenem-resistant bacteria (9).
Comment 7: What about Methicillin and Cefoxitin resistance pattern of Staph. aureus isolates?
Response 7: Thanks for the good comment. We have mentioned Methicillin and Cefoxitin resistant Staph. aureus isolates on page 5 “5.3 Multidrug-resistant microorganisms “section in red marks, and as follows:
Meanwhile, MRSA was isolated in 9.2% cases of SBP during another prospective study in France (10). In another study launched in Greece, SBP due to GPC was found in 55% of cases when patients received quinolone prophylaxis, and MRSA was the most common isolate (8.5%)(11). Regarding nosocomial infections, the prevalence of cefoxitin/methicillin-resistant Staphylococcus aureus (MRSA) was 24.8% in a study launched in France(4). In comparison with non-MRSA infections, MRSA infections were more likely to recur and occurred in more sites other than ascitic fluid and blood (P<0.0004)(4).
On Page 9-10 section 9.1.2: “Currently, vancomycin, the chief antibiotic for infection by MRSA for years, has limitations in terms of in vitro sensitivity, toxicity, and overall patient management. Due to the increase in the mean MICs of vancomycin in clinical isolates of S. aureus, daptomycin could be considered in priority due to its less hepatic metabolism, renal toxicities and myelotoxicity(12). In addition, many novel antibiotics for MDR Gram-positive microorganisms are being developed, but only with appropriate dosing, utilization and careful monitoring for the emergence of antimicrobial resistance can these current antimicrobial drugs continue to treat Gram-positive pathogens in the future(13).”
Comment 8: Other diagnostic markers of SBP not mentioned
Response 8: Thanks for the good comment. We have mentioned it on page 7 in 6.4 section in red mark and as follows:
Several studies showed that either ascites calprotectin or the ratio of calprotectin to total protein has diagnostic and prognostic value for SBP in patients with liver cirrhosis and ascites(14, 15). Calprotectin is a calcium- and zinc-binding protein detected almost exclusively in neutrophils, and its presence in body fluids is proportional to the influx of neutrophils [4].AF calprotectin is related PMNL. This suggests that it may be a suitable candidate for diagnosis of SBP. In addition, the presence of fecal calprotectin quantitatively relates to intestinal neutrophil migration and is therefore considered as a marker of intestinal inflammation and hence a screening tool for SBP(16).
Reference.
- Wiest R, Krag A, Gerbes A. Spontaneous bacterial peritonitis: recent guidelines and beyond. Gut 2012;61:297-310.
- Fernandez J, Acevedo J, Castro M, Garcia O, de Lope CR, Roca D, Pavesi M, et al. Prevalence and risk factors of infections by multiresistant bacteria in cirrhosis: a prospective study. Hepatology 2012;55:1551-1561.
- Shi L, Wu D, Wei L, Liu S, Zhao P, Tu B, Xie Y, et al. Nosocomial and Community-Acquired Spontaneous Bacterial Peritonitis in patients with liver cirrhosis in China: Comparative Microbiology and Therapeutic Implications. Sci Rep 2017;7:46025.
- Campillo B, Richardet JP, Kheo T, Dupeyron C. Nosocomial spontaneous bacterial peritonitis and bacteremia in cirrhotic patients: impact of isolate type on prognosis and characteristics of infection. Clin Infect Dis 2002;35:1-10.
- Fernández J, Navasa M, Gómez J, Colmenero J, Vila J, Arroyo V, Rodés J. Bacterial infections in cirrhosis: epidemiological changes with invasive procedures and norfloxacin prophylaxis. Hepatology 2002;35:140-148.
- Oladimeji AA, Temi AP, Adekunle AE, Taiwo RH, Ayokunle DS. Prevalence of spontaneous bacterial peritonitis in liver cirrhosis with ascites. Pan Afr Med J 2013;15:128.
- Bhuva M, Ganger D, Jensen D. Spontaneous bacterial peritonitis: an update on evaluation, management, and prevention. Am J Med 1994;97:169-175.
- Fernandez J, Navasa M, Gomez J, Colmenero J, Vila J, Arroyo V, Rodes J. Bacterial infections in cirrhosis: epidemiological changes with invasive procedures and norfloxacin prophylaxis. Hepatology 2002;35:140-148.
- Pampalone M, Vitale G, Gruttadauria S, Amico G, Iannolo G, Douradinha B, Mularoni A, et al. Human Amnion-Derived Mesenchymal Stromal Cells: A New Potential Treatment for Carbapenem-Resistant Enterobacterales in Decompensated Cirrhosis. Int J Mol Sci 2022;23.
- Dupeyron C, Campillo SB, Mangeney N, Richardet JP, Leluan G. Carriage of Staphylococcus aureus and of gram-negative bacilli resistant to third-generation cephalosporins in cirrhotic patients: a prospective assessment of hospital-acquired infections. Infect Control Hosp Epidemiol 2001;22:427-432.
- Alexopoulou A, Papadopoulos N, Eliopoulos DG, Alexaki A, Tsiriga A, Toutouza M, Pectasides D. Increasing frequency of gram-positive cocci and gram-negative multidrug-resistant bacteria in spontaneous bacterial peritonitis. Liver Int 2013;33:975-981.
- Falcone M, Russo A, Pacini G, Merli M, Venditti M. Spontaneous Bacterial Peritonitis Due to Methicillin-Resistant Staphylococcus Aureus in a Patient with Cirrhosis: The Potential Role for Daptomycin and Review of the Literature. Infect Dis Rep 2015;7:6127.
- Koulenti D, Xu E, Mok IYS, Song A, Karageorgopoulos DE, Armaganidis A, Lipman J, et al. Novel Antibiotics for Multidrug-Resistant Gram-Positive Microorganisms. Microorganisms 2019;7.
- Weil D, Heurgue-Berlot A, Monnet E, Chassagne S, Cervoni JP, Feron T, Grandvallet C, et al. Accuracy of calprotectin using the Quantum Blue Reader for the diagnosis of spontaneous bacterial peritonitis in liver cirrhosis. Hepatol Res 2019;49:72-81.
- Lutz P, Pfarr K, Nischalke HD, Kramer B, Goeser F, Glassner A, Wolter F, et al. The ratio of calprotectin to total protein as a diagnostic and prognostic marker for spontaneous bacterial peritonitis in patients with liver cirrhosis and ascites. Clin Chem Lab Med 2015;53:2031-2039.
- Gundling F, Schmidtler F, Hapfelmeier A, Schulte B, Schmidt T, Pehl C, Schepp W, et al. Fecal calprotectin is a useful screening parameter for hepatic encephalopathy and spontaneous bacterial peritonitis in cirrhosis. Liver Int 2011;31:1406-1415.

Reviewer 2 Report
- Good work, very detailed and updated.
- Figures 1 and 2 are not displayed if you can kindly insert them separately as attachments.
- suggested reference: https://pubmed.ncbi.nlm.nih.gov/35055040/ Human Amnion-Derived Mesenchymal Stromal Cells: A New Potential Treatment for Carbapenem-Resistant Enterobacterales in Decompensated Cirrhosis. Int J Mol Sci. 2022 Jan 13;23(2):857. doi: 10.3390/ijms23020857. PMID: 35055040; PMCID: PMC8775978.
Author Response
Comment to the Author
Good work, very detailed and updated.
- Figures 1 and 2 are not displayed if you can kindly insert them separately as attachments.
- suggested reference: https://pubmed.ncbi.nlm.nih.gov/35055040/ Human Amnion-Derived Mesenchymal Stromal Cells: A New Potential Treatment for Carbapenem-Resistant Enterobacterales in Decompensated Cirrhosis. Int J Mol Sci. 2022 Jan 13;23(2):857. doi: 10.3390/ijms23020857. PMID: 35055040; PMCID: PMC8775978.
Response 2: Thanks for the good comment. We have displayed Figures 1 and 2 as separate files and added the suggested reference on page 12 “Other novel therapeutic strategies” section and as follows: The emergence of MDR bacteria may require novel therapeutic strategies that do not involve the use of antibiotics. One potential approach is human amniotic mesenchymal stromal cell (hA-MSC) treatment. One in vitro study found hA-MSCs added to ascites fluid could significantly reduce the proliferation of both bacterial strains at 24 h as well as affect M1/M2 polarization, C3a complement protein, and ficolin 3 concentrations during the course of infection in a strain-dependent manner (1). Validation of an in vivo model is warranted for future hA-MSC application in treating ascites infected with carbapenem-resistant bacteria (1).
Reference.
- Pampalone M, Vitale G, Gruttadauria S, Amico G, Iannolo G, Douradinha B, Mularoni A, et al. Human Amnion-Derived Mesenchymal Stromal Cells: A New Potential Treatment for Carbapenem-Resistant Enterobacterales in Decompensated Cirrhosis. Int J Mol Sci 2022;23.

Round 2
Reviewer 1 Report
Dear Author/
Thank you for considering Livers Journal and resubmitting your paper after replying to reviewers comments
Author Response
Response: Thanks for the excellent comment. Based on the studies done by Prof. BRUCE A. RUNYON(1, 2) in the 1980s, we quoted that “percentage of cases with positive cultures to 91%”. But it was a long time ago when prompt antibiotics were not widely used. To avoid disputes and up-to-date, we delete “to 91%” and change the sentence in section 6.3. to “At least 10 mL of AF in both aerobic and anaerobic blood culture bottles can increase the percentage of cases with positive cultures and accelerate the detection time for bacterial growth(1, 2). However, up to 60% of ascites culture may be negative in patients with clinical manifestations suggestive of SBP and increased ascites neutrophil count(3)”.
Reference.
- Runyon BA, Umland ET, Merlin T. Inoculation of blood culture bottles with ascitic fluid. Improved detection of spontaneous bacterial peritonitis. Arch Intern Med 1987;147:73-75.
- Runyon BA, Canawati HN, Akriviadis EA. Optimization of ascitic fluid culture technique. Gastroenterology 1988;95:1351-1355.
- European Association for the Study of the Liver. Electronic address eee, European Association for the Study of the L. EASL Clinical Practice Guidelines for the management of patients with decompensated cirrhosis. J Hepatol 2018;69:406-460.
